# The Effects of Bioactive Compounds from Blueberry and Blackcurrant Powder on Oat Bran Pastes: Enhancing In Vitro Antioxidant Activity and Reducing Reactive Oxygen Species in Lipopolysaccharide-Stimulated Raw264.7 Macrophages

**DOI:** 10.3390/antiox10030388

**Published:** 2021-03-05

**Authors:** Xiao Dan Hui, Gang Wu, Duo Han, Xi Gong, Xi Yang Wu, Shu Ze Tang, Margaret A. Brennan, Charles S. Brennan

**Affiliations:** 1Department of Wine, Food and Molecular Biosciences, Faculty of Agriculture & Life Sciences, Lincoln University, Christchurch 7647, New Zealand; xiaodan.hui@lincolnuni.ac.nz (X.D.H.); gang.wu@lincolnuni.ac.nz (G.W.); xi.gong@lincolnuni.ac.nz (X.G.); margaret.brennan@lincoln.ac.nz (M.A.B.); 2Department of Food Science and Engineering, Jinan University, Guangzhou 510632, China; handuo@stu2018.jnu.edu.cn (D.H.); tkentwu@jnu.edu.cn (X.Y.W.); tangsz@jnu.edu.cn (S.Z.T.); 3Riddet Institute, Palmerston North 4442, New Zealand; 4School of Science, RMIT University, 124 La Trobe Street, Melbourne, VIC 3000, Australia

**Keywords:** antioxidant, in vitro digestion, phenolic compounds, reactive oxygen species, macrophages

## Abstract

In this study, blueberry and blackcurrant powder were chosen as the phenolic-rich enrichments for oat bran. A Rapid Visco Analyser was used to form blueberry and blackcurrant enriched oat pastes. An in vitro digestion process evaluated the changes of phenolic compounds and the in vitro antioxidant potential of extracts of pastes. The anthocyanidin profiles in the extracts were characterised by the pH differential method. The results showed that blueberry and blackcurrant powder significantly increased the content of phenolic compounds and the in vitro antioxidant capacity of pastes, while the total flavonoid content decreased after digestion compared to the undigested samples. Strong correlations between these bioactive compounds and antioxidant values were observed. Lipopolysaccharide-stimulated RAW264.7 macrophages were used to investigate the intracellular antioxidant activity of the extracts from the digested oat bran paste with 25% enrichment of blueberry or blackcurrant powder. The results indicated that the extracts of digested pastes prevented the macrophages from experiencing lipopolysaccharide (LPS)-stimulated intracellular reactive oxygen species accumulation, mainly by the Kelch-like ECH-associated protein 1 (Keap1)/nuclear factor erythroid 2-related factor 2 (Nrf2) signalling pathway. These findings suggest that the bioactive ingredients from blueberry and blackcurrant powder enhanced the in vitro and intracellular antioxidant capacity of oat bran pastes, and these enriched pastes have the potential to be utilised in the development of the functional foods.

## 1. Introduction

Oxidative stress plays a central role in the initiation and progression of several chronic diseases, including obesity and diabetes. Oxidative stress can damage cellular structures along with the under-production of antioxidant mechanisms, leading to the progression of obese or diabetic-related complications [1,2]. The ensuing cellular damage, such as DNA cross-linking and apoptosis has been reported to be a result of oxidative stress, and it is a fundamental pathological process in a variety of chronic diseases [3]. These diseases have been shown to possess increased cellular levels of reactive oxygen species (ROS) and ROS-induced DNA damage [4]. The Kelch-like ECH-associated protein 1 (KEAP1)/nuclear factor erythroid 2-related factor 2 (Nrf2) stress response pathway is the principal inducible defence against oxidative stress [5]. This signalling pathway regulates the expression of more than 100 genes and functions related to oxidative stress and cell survival, including direct antioxidant proteins, electrophile detoxification enzymes, free radical metabolism, the recognition of DNA damage, and the inhibition of inflammation [6,7].

With the growing number of people being diagnosed with these metabolic syndromes, it is crucial to find a new solution. Epidemiological studies and randomised control trials [8] have shown that dietary modification (in particular whole fruit and grains intake) are protective against diabetes and obesity since the complex mixture of phytochemicals from these foods has additive and synergistic effects [9]. Therefore, the consumption of antioxidant molecules has been shown to be effective as a strategy to prevent, or reduce, the risk of these diseases.

Research [10] has indicated the benefits of natural antioxidants, such as polyphenols derived from natural plants, compared with synthetic antioxidants. However, the intensified accumulation, safe consumption, and toxic effects of these polyphenols should be considered [11]. Polyphenols have been studied in cell culture and animal studies for their protective role. Various coloured berries, especially blueberry and blackcurrant—widely distributed in New Zealand [12]—contain a large number of polyphenols, such as phenolic acids and flavonoids [13]. Anthocyanin is a major subclass of flavonoids [14]. The major anthocyanins identified in blueberries are 3-glycosidic derivatives of cyanidin, delphinidin, and malvidin, in which the most common derivatives detected are based on sugars, such as glucose [15]. The major anthocyanins in blackcurrant are delphinidin-3-O-glucoside, cyanidin-3-O-glucoside, and cyanidin-3-O-rutinoside [16,17]. These bioactive compounds have been found to protect cells from oxidative stress [18] and to improve the ability of plasma antioxidants, thus reducing the risk of different human chronic diseases, including obesity and type 2 diabetes [19,20]. This increase in plasma antioxidant capacity following the consumption of polyphenol-rich food may be responsible either for the presence of the metabolites of polyphenols in plasma, or for their preservative effects on other reducing agents, such as endogenous antioxidants, or for their effect on the absorption of pro-oxidative food components, such as iron [21].

A whole grain oat diet has been credited in conferring health-promoting benefits. Oat bran is particularly high in antioxidants compared to other parts of the oat grain [22]. Recent studies have also shown that the health benefits of oats are mainly due to the antioxidants found in the bran—in addition to the phenolic compounds, such as potent avenanthramides, which are a family of antioxidants unique to oats [23]. Avenanthramides are substituted N-cinnamoyl anthranilic acids, consisting of anthranilic acid and cinnamic acid moieties. These bioactive compounds in oat bran have been demonstrated to have the potential to reduce inflammation, possess anticancer properties, and lower blood sugar levels [24]. A concentrated extract of oat bran can be used as a natural antioxidant for foods, protecting the long-chain fatty acids from oxidative stress and from creating off-flavours in foods, since oat bran is a good source of antioxidants [25].

Consumer’s interest in naturally coloured foods is growing. Free-flowing dried fruit powder can be easily incorporated into foods in a mixed form. Dehydrated fruit powders, such as blueberry and blackcurrant, can be mixed with oatmeal to prepare breakfast cereals [20,26,27]. A study of Schmidt, et al. [28] revealed that dried wild blueberry powder did not decrease the in vitro anti-proliferation activity in comparison to that of the fresh fruit. However, few studies have reported the use of combinations of berry fruits with oat bran, and no studies have focused on the potential synergistic effect on the food matrix of combining berries and oat bran. Therefore, in this study, blueberry powder and blackcurrant powder were chosen as the phenolic-rich enrichments for the oat bran. An in vitro digestion process was performed to observe the food matrix effects on the changes in their phenolic contents and in vitro antioxidant potential. The pH differential method was conducted to identify the major anthocyanidins in blueberry and blackcurrant enriched pastes. Furthermore, lipopolysaccharide-stimulated RAW264.7 macrophages were employed to investigate the intracellular antioxidant activity of the extracts from blueberry and blackcurrant enriched pastes. The potential mechanisms of this intracellular activity elicited by the extracts were also studied.

## 2. Materials and Methods

### 2.1. Chemicals and Materials

Blueberry powder and blackcurrant powder were purchased online (Viberi, Timaru, New Zealand). Oat bran was obtained from the local supermarket (Sun Valley, Christchurch, New Zealand). Pepsin (EC 3.4.23.1) pancreatin (EC 232-468-9), α-amyloglucosidase (EC 3.2.1.3), invertase (EC 3.2.1.26), 2,2′-Azino-bis (3-ethylbenzothiazoline-6-sulfonic acid) diammonium salt (ABTS), 2,4,6-Tris(2-pyridyl)-s-triazine (TPTZ), 2-Diphenyl-1-picrylhydrazyl (DPPH), 3,5-Dinitrosalicylic acid (DNS, 98%, ACROS Organics™, Waltham, MA, USA ), Folin and Ciocalteu’s phenol reagent, 2,7 dichlorodihydrofluorescein-diacetate (DCFH-DA), gallic acid, rutin, and trolox (6-hydroxy-2,5,7,8-tetramethylchroman-2-carboxylic acid) were all purchased from Sigma-Aldrich (St. Louis, MO, USA). Other chemicals in this study were of analytical grade.

### 2.2. Preparation of the Pastes

Oat bran (OB), blueberry (BB) powder and blackcurrant (BC) powder were the raw materials used for the preparation of food matrices. Pastes were made using a Rapid Visco Analyser (RVA-Super 4, Perten instruments, Sydney Australia) [29]. The equations (1) and (2) were used to determine the weight of water and raw materials. OB (5.28 g, 2.34%, moisture basis) supplemented with 0, 10, 15, and 25% (w/w) BB powder or BC powder was prepared to develop the pastes. The slurry was heated from 50 to 95 °C at a rate of 6 °C/min, held at 95 °C for 5 min, then cooled at a rate of 6 °C/min to 50 °C, and finally held at 50 °C for 2 min. The spindle speed was kept at 160 rpm, except for the first 10 s where it was increased to 960 rpm to disperse the mixture. The pastes were coded as OBP (nothing added, control group); ABB_10_, ABB_15_, and ABB_25_ (oat bran paste enriched with 10%, 15%, and 25% BB powder, respectively), and ABC_10_, ABC_15_, and ABC_25_ (oat bran paste enriched with 10%, 15%, and 25% BC powder, respectively). All the pastes were stored at 4 °C overnight.

Equivalent sample and water mass can be calculated using the following formulas [30]. This is normalised for the moisture of the oat flour at 14% (gives a factor of 86 in the equation):$$S\;=\;\frac{86\;\times \;6.0}{100\;-\;M}$$
W = 25 + (6.0 − S)
where *S* = corrected sample weight (g), *W* = corrected water weight (g) and *M* = actual moisture content of sample (in %).

### 2.3. Extraction of Phenolics from Raw Materials and Pastes

All the pastes were freeze dried at −30 to −40 °C using pilot scale lyophilization system (Millrock Technology, Inc., Kingston, NY, USA) for 72 h, and then the lyophilised samples were put in vacuum-incubators, and stored at 4 °C. Afterwards, the extraction procedure was performed with two solvent systems [31]: acidic methanol/water (50:50 *v*/*v*, pH = 2) and acetone/water (70:30 *v*/*v*, pH = 2), respectively, followed with the step of vortexing, sonication, centrifugation and evaporation. The final extracts were stored at −80 °C.

### 2.4. Simulation of the In Vitro Digestion

The in vitro digestion method was modified according to a previous study [32]. The entire procedure was performed in a 37 °C incubator with constant shaking table at 120 r/min. A total of 2 g of each lyophilised sample was mixed well with 30 mL of distilled water for 10 min, and then the pH value of the mixture was adjusted to 2.0 with 6 N HCl. Pepsin was added at a concentration of 0.05 g/mL of the sample, and the mixture was incubated for 1 h. After finishing the gastric digestion, a 1 mL of aliquot from each sample was taken (time 0) and added to 4 mL of absolute ethanol to stop further reactions. The pH of the digest was adjusted to 6.0 by the dropwise addition of 0.9 M NaHCO_3_. After the pH adjustment, 0.1 mL of α-amyloglucosidase (3000 U/mL) was added. The digestion time began as soon as 5 mL of pancreatin-bile solution (3 g/mL pancreatin and 0.025 g/mL bile salts in 0.1 M NaHCO_3_, pH = 7.4) was added. After incubating for 120 min, aliquots from the digesta of each sample were individually treated with ethanol and centrifuged at 2500 g for 20 min. The supernatants were collected. The digested samples were extracted, followed by the extraction procedure in Section 2.3, and then filtered through 0.22 μm Millipore filters and stored at −80 °C for further analysis.

### 2.5. Determination of the Total Phenol Content (TPC)

The total phenol content (TPC) of the extracts from undigested and digested samples was measured as described by Kim and Lee [33], with some modifications. A total of 0.5 mL of each sample was placed in tubes and 2.5 mL of 0.2 N Folin–Ciocalteu reagent and 2.0 mL of 7.5% Na_2_CO_3_ were added to each tube. These tubes were mixed well and incubated in a water bath at 40 °C for 30 min. Once the mixture was cooled to room temperature, the absorbance was measured at 760 nm by a spectrophotometer. Gallic acid was used as a standard to determine TPC of the extract and digesta as mg gallic acid equivalent (GAE)/g sample.

### 2.6. Determination of the Total Flavonoid Content (TFC)

Total flavonoid content (TFC) was measured using aluminium chloride reagent [34]. A total of 250 μL of each sample was mixed with 75 μL of sodium nitrite solution (5%, *w*/*v*), followed by 150 μL of aluminium chloride (10%, *w*/*v*), 500 μL of sodium hydroxide (1 mol/L), and finally, 775 μL of distilled water. The mixture was shaken and incubated at room temperature for 30 min. The absorbance of the mixture was measured at 415 nm. Results are expressed as mg rutin equivalents (RE)/g sample.

### 2.7. Determination of the Total Monomeric Anthocyanins Content (TMAC)

Total monomeric anthocyanins content (TMAC) was determined by the pH differential method [35]. Extracts were diluted separately with 0.025 mol/L hydrochloric acid–potassium chloride buffer (pH = 1) and 0.4 mol/L sodium acetate buffer (pH = 4.5). These dilutions were allowed to balance for 15 min. The absorbance of the mixture was measured at 530 nm and 700 nm, respectively, using a UV–Vis spectrophotometer (UV1800, Shimadzu, Kyoto, Japan). The absorbance of the diluted sample was calculated according to the equation:
*A* = (*A*_530 nm_ − *A*_700 nm_)_pH1.0_ − (*A*_530 nm_ − *A*_700 nm_)_pH4.5_

The TMAC was expressed as mg cyanidin-3-glucoside equivalents (Cy-3GE)/g sample as in the equation:$$\mathrm{Anthocyanidin}\;\mathrm{pigment}\;(\mathrm{mg}/L)\;=\;\frac{A\;\times \;MW\;\times \;DF\;\times V\;\times \;1000}{\varepsilon \;\times \;l\;\times \;m}$$
where *A* is the absorbance, *MW* is the molecular weight of cyanidin-3-glucoside (449.2 g/mol), *DF* is the dilution factor, *V* is the solvent volume (mL), *ε* is the molar absorptivity (26,900 L·mol^−1^·cm^−1^) [36], *l* is the cell path length (1 cm) and *m* is the sample weight.

### 2.8. In Vitro Antioxidant Activity Assay

#### 2.8.1. DPPH Assay

The radical scavenging capacity of the extract was determined by the DPPH assay, as described by Floegel et al. [37]. A total of 1 mL of freshly prepared 0.1 mM methanolic DPPH solution was added into 0.5 mL of the extract, or digesta, and incubated for 30 min in the dark. The absorbance of the reaction mixture was measured at 517 nm. Trolox was used as standard and the DPPH radical scavenging capacity was expressed as μmoL trolox equivalent (TE)/g sample.

#### 2.8.2. ABTS Assay

The ABTS assay was adapted from a previous study [37]. The ABTS^·+^ cation radical solution was produced by reacting 9.5 mL of 7 mM ABTS stock solution and 245 μL of 100 mM K_2_S_2_O_8_ solution and incubating this solution in the dark at room temperature for 16 h before use. The ABTS^·+^ radical cation solution was diluted with PBS (pH = 7.4) to an absorbance of 0.70 ± 0.02 at 734 nm. The diluted ABTS^·+^ radical cation solution (3 mL) was thoroughly mixed with 0.3 mL extract or digesta. The mixture was kept in the dark for 6 min at room temperature. Absorbance values were measured at 734 nm. Trolox was used as the standard. Results are expressed as μmoL trolox equivalent (TE)/g sample.

#### 2.8.3. FRAP Assay

The reducing ability and antioxidant power activity of each extract was determined using a Ferric Reducing Antioxidant Power (FRAP) reagent solution [38]. The fresh FRAP reagent solution was prepared with 300 μmol/L acetate buffer (pH = 3.6), 10 mmol/L TPTZ (dissolved in 40 mmol/L HCl) and 20 mM FeCl_3_ at a ratio of 10:1:1 (*v*/*v*/*v*). A total of 2.5 mL of FRAP reagent solution was thoroughly mixed with 250 μL extract. The mixture was incubated in the dark for 2 h at 37 °C and the absorbance was measured at 593 nm. FeSO_4_ solution was used as the standard. Results are expressed as μmoL Fe^3+^ equivalent (Fe^3+^ E)/g sample.

### 2.9. Cell Culture

The RAW264.7 macrophage was purchased from the Cell Bank of the Shanghai Institute of Cell Biology and Biochemistry, Chinese Academy of Sciences (Shanghai, China). The cells were maintained in DMEM containing 100 U/mL penicillin, 100 μg/mL streptomycin, 2 mmol/L glutamine, and 10% fetal bovine serum. The cell was cultured at 37 °C a humidified atmosphere of 5% CO_2_.

### 2.10. Determination of Cell Viability

The intestinal digested extracts of OBP, ABB_25_, and ABC_25_ were selected to study the effects of bioactive compounds from the extracts of pastes on the intracellular antioxidant activity of RAW264.7 macrophages. RAW264.7 macrophages were treated with digested extracts (OBP, ABB_25_, and ABC_25_) individually and plated at a density of 3.0 × 10^4^ cells/well in 96-well culture plates for 48 h. Cell viability was determined using a Cell Counting Kit-8 (CCK-8) assay kit (Dalian Meilun Biotechnology Co., Ltd., Dalian, China) according to the instructions of the manufacturer. Absorbance was calculated for all samples at 450 nm (OD_450_). The relative cell viability was presented after normalised to untreated cells (control). Cell viability rates were measured after 24 h and were calculated based on OD_450_ values. Cell viability rate (%) = OD_450_ (test)/OD_450_ (control) × 100%.

### 2.11. Induction of Intracellular ROS Generation

The intracellular changes in ROS generation were detected by staining the cells with 2,7 dichlorodihydrofluorescein-diacetate (DCFH-DA) [39]. RAW264.7 macrophages were seeded at a density of 4.0 × 10^5^ cells/well in a 12-well culture plate, and were treated with extracts of OBP, ABB_25_, and ABC_25_ (100 and 200 μg/mL) for 24 h, and then 10 μL of lipopolysaccharide (LPS, 250 ng/mL) was added followed by 30 min incubattion. The cells were harvested and washed twice with cold PBS, then, cells were further incubated with 10 μM DCFH-DA at 37 °C for 30 min. Subsequently, the cells were washed two times using PBS. Prior to ROS measurement, 100 μL of PBS was added to each well. ROS generation was assessed by flow cytometry.

### 2.12. Luciferase Reporter Nrf2 Gene Assay

RAW264.7 cells were seeded at a density of 2 × 10^5^ cells/well in a 24-well plate in serum free DMEM (did not contain antibiotics), and incubated at 37 °C for 5 h. Cells were then transfected with Nrf2 using lipofectamine 2000 transfection reagent (invitrogen) and incubated for a further 4 h [40]. Afterwards, cells were treated with the extracts OBP, ABB_25_, and ABC_25_ (100 and 200 μg/mL) for 24 h, and then 10 μL of LPS (250 ng/mL) was added to incubate for 30 min. Luciferase activities were measured using Dual-Glo luciferase assay kit (Promega, Southampton, UK) according to the manufacturer’s instructions.

### 2.13. Western Blotting Assay

After treatments with extracts individually for 24 h, the cells were harvested, collected as cell pellets, and lysed in RIPA cell lysis buffer on ice for 1 h. Protein concentrations were determined using a BCA Protein Assay Kit (Thermo, MA, USA). Equal proteins from each treatment were separated on a 10% SDS denaturing polyacrylamide gel and electrophoretically (SDS-PAGE) transferred to PVDF membranes. After blocking with 5% non-fat milk, the membranes were incubated with primary antibodies (1:1000; Cell Signalling Technology) overnight at 4 °C. Specific primary antibodies against Keap1, Nrf2, HO1 and β-actin were purchased from Beyotime (Shanghai, China). After washing thrice (10 min for each) with TBS solution, the PVDF membranes were incubated with corresponding secondary antibodies (Jackson ImmunoResearch Laboratories, West Grove, USA) for 1 h. The blots were washed thrice (10 min for each) with TBS solution. Signals were detected by using an Enhanced Chemiluminescence (ECL) detection (Thermo, MA, USA) and Image J (Bethesda, MD, USA) software were used to quantify the blot density [40].

### 2.14. Statistical Analysis

The results are presented as the mean value ± standard deviation. Unless stated elsewhere, experiments were performed in triplicate. One-way analysis of variance (ANOVA) was carried out. Pearson’s correlation was conducted by using GraphPad Prism software version 8.0 (GraphPad Software, Inc., San Diego, CA, USA).

## 3. Results

### 3.1. Changes in TPC and TFC during In Vitro Digestion

Table 1 shows the TPC in extracts of raw materials and pastes at different digestion phases. Overall, although there was a decline in the TPC content of both raw materials and pastes after intestinal digestion, compared with the TPC after gastric digestion, the polyphenol concentration of each digested sample was higher than that of the corresponding undigested sample (*p* < 0.01 or *p* < 0.05). Before digestion, BC powder yielded the highest TPC value of 97.15 mg GAE/g, followed by BB powder (84.91 mg GAE/g), while OBP showed the lowest TPC value of 0.43 mg GAE/g. All of the BC-enriched pastes had higher TPC values than those of the BB-enriched pastes. The difference became more evident as the level of BB or BC powder increased in the pastes. The TPC in the extract of undigested ABC_25_ was 10-fold higher than that of undigested ABB_25_ and the ratio reduced to 2.4-fold after the intestinal digestion. Thus, ABC_25_ had the highest TPC value among all pastes (*p* < 0.01).

The TFC of the samples illustrated that BB powder displayed the highest TFC across the whole digestion, while OB had the lowest TFC value. The TFC value of all samples firstly increased after the gastric digestion, then decreased after the intestinal digestion. In OB, the TFC value after the intestinal digestion was so low that it could not be detected. BB-enriched pastes exhibited higher levels of TFC than those of BC-enriched pastes. For instance, before digestion, the TFC value of ABB_25_ was 45.12% higher than that of ABC_25_ (*p* < 0.01). After intestinal digestion, the TFC of ABB_25_ was 30% lower than the corresponding undigested sample, although it was still 9.4% higher than that of the intestinal digested extracts of ABC_25_ (*p* < 0.01). The TFC trend of BB powder > BC powder > BB-enriched pastes > BC-enriched pastes > OB > OBP was maintained for TFC throughout the whole digestion process.

Research has illustrated the instability of TPC in a simulated digestion [41]. Herein, Both raw materials and pastes displayed increased TPC after the gastric phase, which subsequently declined after the intestinal phase, albeit above the undigested levels. Cebeci and Şahin-Yeşilçubuk [42] reported that combinations with milk generally resulted in a decrease in TPC and TFC as well as inhibition of antioxidant activities when evaluating the matrix effect of blueberry, oatmeal and milk on their polyphenols and antioxidant activities after in vitro digestion. This discrepancy might be related to differences in the food matrix characteristics and the in vitro conditions of the digestion. However, Sengul, Surek and Nilufer-Erdil [32] observed a higher recovery of TPC after the gastric digestion of the fruit extract. This finding was possibly due to an increase in flavylium cations in the acidic solution during the gastric phase of digestion, which is in agreement with results from this study. Therefore, it can be assumed that the increased values of TPC and TFC during the gastric phase are due to the acidic hydrolysis of phenolic glycosides to their aglycones. Furthermore, the decline in TFC values, subsequently resulted in the decreased TPC values in the intestinal phase. This is attributed to the degradation of phenolic compounds in the weak alkaline environment (pH = 7.4), particularly flavonoids, which are highly sensitive to alkaline conditions. According to the study from Fawole and Opara [43] the decrease in phenolic compounds—notably anthocyanins in the intestinal phase of the in vitro digestion—was attributed to the transformation of the flavylium cation to the colourless chalcone when the digestion medium became alkaline.

Changes in TPC and TFC of the extracts were influenced by the combined effects of dietary supplements and the phases of in vitro digestion, and the effects were considered extremely significant (*p* < 0.01). Crucially, these results show that the changes in TPC in extracts from raw materials and pastes were not constant throughout the in vitro digestion procedure. According to the percentage of total variation, for TPC and TFC, the digestion phase accounted for the most significant variation, indicating that the digestion process most likely contributes to the release of bioactive compounds due to the actions of the digestive enzymes, temperature, and pH conditions [44].

### 3.2. The Changes in TMAC of the Extracts during In Vitro Digestion

The anthocyanin pigment experiences a reversible structural transformation with changes of the pH value, and these changes are reflected in the absorption spectrum. Thus, the pH differential method could measure the values rapidly and accurately, even when the degraded polymerised pigments and other interfering compounds were present [15]. Table 2 illustrates the changes in the TMAC values in the extracts of raw materials and pastes during in vitro digestion. Amongst the extracts from undigested samples, the extract of BC powder had the highest TMAC (36.27 mg Cy-3GE/100 g), followed by the extract of BB powder (14.96 mg Cy-3GE/100 g). TMAC in the OB extract was low, 0.03 mg Cy-3GE/100 g. In terms of the extracts from pastes, the TMAC in the extract of OBP was too low to be detected. TMAC in the extracts of BB- and BC-enriched pastes varied from 0.25 to 0.66 mg Cy-3GE/100 g. The extract of BC-enriched paste was given more TMAC compared to the corresponding extract of BB-enriched paste (*p* < 0.01). TMAC in all extracts decreased after undergoing in vitro digestion. TMAC in the extracts of BB and BC powder decreased by over 95% compared to the corresponding undigested extracts (*p* < 0.01). TMAC in the OB extract could not be detected in both the gastric and intestinal phase. Even though the TMAC in the extracts from the pastes experienced a decrease after in vitro digestion, BC-enriched pastes still exhibited higher TMAC values compared with the corresponding BB-enriched pastes. Among these extracts from intestinal digested pastes, ABC_25_ had the highest TMAC value (0.35 mg Cy-3GE/100 g) (*p* < 0.01).

The instability of these extracts could be responsible for the change of the pH values during the digestion. As a food colorant, anthocyanin is very sensitive to a higher pH (alkaline conditions). Therefore, the decline in the TMAC values after the intestinal phase was less pronounced than the decrease after the gastric phase. In addition to the influence of pH values, the temperature may have had a significant effect on the degradation of anthocyanins. Refrigeration has been shown to be an effective means of preserving anthocyanins. Muche et al. [45] compared the amount of anthocyanidin content lost in blackcurrant juice stored at 4 and 37 °C, observing that blackcurrant anthocyanidin contents lost 40% at 4 °C, while no measurable amounts of blackcurrant anthocyanidin were found at 37 °C. The short-term high temperature treatment could improve the stability of anthocyanin by facilitating the inactivation of native enzymes that are harmful to anthocyanins. Herein, the decline in the TMAC in the extracts of the pastes was less than those in the extracts of raw materials. This could because during the formation of the pastes, samples experienced a short-term high temperature procedure (95 °C for 10 min), which may have prevented anthocyanins from degrading to some extent [46].

In addition, the freeze-drying technique is widely used in high-quality food preparation, as this technique has been proven to be able to preserve the bioactive compounds effectively [47]. Therefore, in this study, all the pastes were treated with freeze-drying for 72 h to preserve the bioactive ingredients in these pastes.

### 3.3. Changes in Antioxidant Activity during In Vitro Digestion

The radical scavenging activity of the samples prior to digestion, as measured by scavenging DPPH and ABTS free radicals, reflected the same trend as the TPC levels in all samples, with BC powder > BB powder > BC-enriched pastes > BB-enriched pastes > OB > OBP (Figure 1a,b). After the gastric phase, the radical scavenging activity of both raw materials and pastes decreased, varying from a 13 to 33% reduction in the DPPH assay and from a 20 to 89% reduction in the ABTS assay (*p* < 0.01). The radical scavenging activity of all raw materials after the intestinal phase was lower than that of the corresponding undigested extracts. Both BB- and BC-enriched pastes seemed to be effective in scavenging the DPPH and ABTS^·+^ free radical, and contrary to the raw materials, the radical scavenging activity of these pastes was higher after the intestinal digestion compared to the radical scavenging activity exhibited by the undigested pastes (*p* < 0.01). The DPPH values of the intestinal digesta of BB- and BC-enriched pastes increased between 1- and 1.54-fold compared with the undigested extracts (*p* < 0.01). The same trend was observed for the ABTS assay results, with the scavenging values of intestinally digested BB- and BC-enriched pastes being higher by 14 to 53%, compared to the corresponding undigested pastes (*p* < 0.01).

The reducing antioxidant powers measured by FRAP of the raw materials and pastes were consistent with the TPC measured in the undigested paste (Figure 1c). Overall, BC powder showed the highest antioxidant power (714 mmol FeSO_4_/g), followed by BB powder (282 mmol FeSO_4_/g). In terms of the pastes, OBP showed the lowest antioxidant power (13 mmol FeSO_4_/g). With increasing levels of enrichments, the FRAP value of BB- and BC-enriched pastes increased, ranging between 0.9- and 6.2-fold compared with OBP (*p* < 0.01). BC-enriched pastes showed stronger reducing power than BB-enriched pastes. Interestingly, the FRAP values increased significantly at the gastric phase for all samples, perhaps as a result of the TPC at this phase. However, the reducing powers then generally decreased (*p* < 0.01) by 7 to 76 % at the intestinal phase, with larger decreases observed in the raw materials values compared with those observed for the pastes. Although there was a decrease after the final phase, FRAP values after the intestinal digestion remained higher than the undigested extracts (*p* < 0.01).

The fluctuation of the reducing power, in all of the samples, could be due to the pH of the medium. The pH of a substance is known to influence molecule racemisation, possibly creating two chiral enantiomers with different reactivities. Therefore, some antioxidants could be more sensitive at acidic pH in the gastric phase and less reactive at alkaline pH in the intestinal phase. Bouayed, et al. [48] reported a similar trend of reducing powder in apple varieties. They found that the reducing antioxidant capacity present in apple varieties, as determined in methanolic extracts, was significantly higher compared to those found in gastric digesta for all apple varieties, while the reducing power present in the intestinal digesta of apple varieties was lower than those found in gastric digesta. Regarding the change in reduction of antioxidant power from the gastric phase to the intestinal phase, the phenolics responsible for ferric reduction may reduce or convert to certain metabolites with different chemical properties, as these polyphenols are highly sensitive to alkaline conditions [49].

The DPPH and ABTS assays are based on the ability of antioxidants to scavenge the DPPH and ABTS^·+^ radicals activities. The DPPH and ABTS values of the pastes showed a different behaviour from the FRAP value, displaying a decline in the gastric phase, followed by an increase after the intestinal phase in relation to the gastric phase. The decline in DPPH and ABTS values in the gastric phase could be related to the degradation of anthocyanins responsible for the scavenging activities due to the acidic environment. The increased DPPH and ABTS values could be to the deprotonation of the hydroxyl groups present on the aromatic rings, facilitating the hydrogen donation reactions and decreasing the ionization potential, consequently increasing the electron donation capacity. In addition, the formation of new metabolites from anthocyanins that can become antioxidant radicals could lead to the increased DPPH and ABTS values.

Çelik, et al. [50] reported that once cereal products are consumed, a large proportion of insoluble antioxidants bound to dietary fibres reach the colon, without digestion occurring, and become bound to antioxidant radicals themselves. Meanwhile, soluble antioxidants regenerate bound antioxidant radicals, thereby prolonging their antioxidant action. Masisi et al. [51] also reported that after the consumption of breakfast cereals, antioxidant activities increased significantly after in vitro digestion compared to the chemical solvent extraction procedure used for undigested samples. These reports explain the increased reducing power of all blueberry and blackcurrant enriched pastes. Antioxidant activities of foods varies depending on the content of phenolic compounds, flavonoids, proteins, lipids and carbohydrates [52].

### 3.4. Pearson’s Correlations between Phenolic Compounds and Three Antioxidant Assays

The correlation coefficients between TPC, TFC, and the anthocyanidin content (Table 3) and antioxidant activity values—analysed by the three assays—were recorded before and after in vitro digestion. Strong positive correlations were observed between these phenolic compounds and antioxidant activity values. The anthocyanidin content had stronger correlations with antioxidant activity values compared to TPC and TFC. As shown in Table 3, the correlation coefficients between the anthocyanin content and DPPH values were all larger than 0.97 in the undigested (R^2^ = 0.992, *p* < 0.01), gastric digested (R^2^ = 0.982, *p* < 0.01), and digested samples (R^2^ = 0.979, *p* < 0.01), followed by the FRAP values, which had stronger correlations in the undigested and gastric digested samples (R^2^ = 0.996, *p* < 0.01) when compared to the intestinal digested samples. In our previous study [53], four types of anthocyanidins, including delphinidin, cyanidin, petunidin, and malvidin, were identified in the extract of blueberry powder-enriched paste, while delphinidin and cyanidin were detected in the extract of blackcurrant powder-enriched paste. Therefore, these anthocyanidins could be responsible for the radical scavenging activity and reducing power in the sample extracts. However, the functionality of the food is not only dependent on the bioactive compounds in food system, but is also highly influenced by the other surrounding compounds, such as lipids, proteins and fibres. Therefore, the whole food matrix should be taken into consideration. In our previous study, the nutritional components in the raw materials and pastes, and the negative correlations were observed between antioxidant activity (DPPH, ABTS^·+^ and FRAP) and nutritional components, including fat, protein and total starch content, while strong positive correlations were found between antioxidant activity and p-coumaric acid content and garlic acid content. Previously, metabolites formed as a result of structural changes brought about by the alkaline conditions have been shown to have a different reactivity in the FRAP assay [54]. However, some insoluble antioxidant compounds remaining in indigestible materials may be underestimated by the three assays [55].

### 3.5. The Extracts of BB- and BC-Enriched Pastes Reduced the Intracellular ROS Level in LPS-Stimulated RAW264.7

A CCK-8 assay was performed to evaluate the effects of the extracts from intestinal digested blueberry and blackcurrant enriched pastes on RAW264.7 cell viability. As shown in Figure 2, the extracts of digested 25% blueberry and 25% blackcurrant enriched pastes exhibited stronger inhibitory activity against cell growth of RAW264.7, compared to the extract of digested OBP (*p* < 0.01). In order to make sure the cell viability was more than 90%, the concentrations of 100 and 200 µg/mL extracts of digested pastes were chosen for subsequent analyses.

LPS-induced intracellular ROS accumulation was monitored within cells using a DCFH2-DA fluorescence microscopic analysis. As shown in Figure 3, the incubation of LPS-stimulated RAW264.7 cells with the extract of OBP (100 and 200 µg/mL) showed no significant reduction in the intracellular ROS, while co-treatment with the extracts of BB- and BC-enriched pastes (100 and 200 µg/mL) resulted in a considerable (*p* < 0.05) dose-dependent reduction in ROS accumulation in LPS-induced RAW264.7 cells. Among these, the 200 µg/mL extract of ABC_25_ exhibited the most significant reduction of ROS concentration, reducing the ROS level by approximately four-fold compared to the control group (*p* < 0.01). Therefore, it can be suggested that the enrichment of BB or BC powder for oat bran could suppress LPS-induced ROS generation in macrophages.

ROS are created by a variety of cellular processes as part of cellular signalling events. A number of studies have suggested that ROS participate in inflammation, and LPS-induced ROS generation has been widely reported in various in vitro and in vivo studies [56]. In particular, ROS are major oxidative products that are primarily released by the mitochondria, peroxisomes, and inflammatory cell activation by endotoxins in macrophages. Moreover, ROS production is an important component of the initiation and enhancement of cell death via apoptosis or autophagy [57]. In this study, stimulation of LPS enhanced intracellular ROS accumulation in RAW264.7 macrophages, and this elevated intracellular ROS accumulation was significantly inhibited by the extracts of BB- and BC-enriched pastes (bioactive compounds from BB and BC powder).

### 3.6. The Extracts of BB- and BC-Enriched Pastes Activated the Antioxidant Gene Nrf2 in LPS-Stimulated RAW264.7 via Keap1/Nrf2/HO-1 Signalling Pathway

Nrf2 is a transcription factor that serves as a sensor for oxidative stress, and coordinates the expression of antioxidant stress response genes upon exposure to oxidative stimulation [58]. Herein, we hypothesised that the protective effects of the extracts of digested BB- and BC-enriched pastes against LPS-stimulated oxidative stress (intracellular ROS accumulation) were regulated by the induction of antioxidant genes through the Nrf2 gene. The results of the luciferase reporter gene assay (Figure 4) showed that the extracts of BB- and BC-enriched pastes (100 and 200 µg/mL) significantly increased the expression of the Nrf2 gene compared to the extract of OBP (*p* < 0.05). Among these extracts, the extract of 200 µg/mL ABC_25_ led to the most significant increase in the expression of the Nrf2 gene. This result is in agreement with the concentration of intracellular ROS accumulation in RAW264.7 macrophages. Collectively, these findings suggest that the bioactive compounds from BB and BC powder could promote the activation of the Nrf2 gene in macrophages.

To further confirm this protective effect, the extracts of ABB_25_ (62.5, 125, and 250 µg/mL) and ABC_25_ (50, 100, and 200 µg/mL) were selected to perform a Western blotting assay to investigate the mechanism of the activation of the Nrf2 gene by the extracts. As shown in Figure 5, the extracts of ABB_25_ and ABC_25_ significantly activated the Nrf2 signalling pathway by down-regulation of the protein expressions of Keap1, and up-regulation of the protein expression of Nrf2 and its downstream factor heme oxygenase 1 (HO1) in a dose-dependent manner (*p* < 0.05).

The Keap1/Nrf2 pathway is the main cytoprotective regulator in responses to endogenous and exogenous stresses induced by ROS [59]. Transcription factor Nrf2, a key signalling protein within the pathway, binds with small Maf proteins to the antioxidant response element (ARE) in the regulatory regions of target genes. Keap1 is a repressor protein that binds to Nrf2 and promotes its degradation via the ubiquitin proteasome pathway [60]. It has been documented that Keap1/Nfr2-mediated antioxidant genes are stimulated by various external stimuli and by plant-derived natural products [61]. Activated Nrf2 binds the DNA at ARE binding motifs to activate the transcription of various detoxifying enzymes, including HO-1 [62]. HO-1 is a well-known antioxidant enzyme which plays a major in the defence against LPS-stimulated ROS generation in macrophages. HO-1 expression has also been shown to be driven by Nrf2 [63]. These antioxidant enzymes are important in the prevention of cell carcinogenesis, and in the protection from oxidative stress and electrophile toxicity. The induction of these antioxidant genes has been assumed to be the mechanism through which Nrf2 inhibits LPS-stimulated inflammation [64]. A previous study [65] revealed that induction of antioxidant enzyme (HO-1) expression can reduce intracellular ROS levels, creating an improved intracellular environment, and maintaining Nrf2 in its augmented configuration. In this study, the expression of the Nrf2 gene in macrophages significantly increased in the presence of the extracts of ABB_25_ and ABC_25_, suggesting that the elimination of intercellular ROS may be due to the activation of Nrf2 by the extracts. This is consistent with previous studies, revealing that the extracts of many plants contain anti-inflammatory agents inhibited the intracellular ROS accumulation through the activation of Nrf2 cascades in macrophages. These results demonstrate that the protective effect of the extracts is mediated primarily by Nrf2 activation. Thus, the bioactive compounds in the extracts protected the macrophages from LPS-stimulated intracellular ROS accumulation, mainly by elevating the intracellular anti-oxidative enzymes via enhancing the accumulation of Nrf2, and thus, dramatically inducing the expression of the antioxidant gene HO-1, in response to LPS stimulation.

## 4. Conclusions

In conclusion, incorporation of 10, 15, and 25% of blackcurrant or blueberry powder with oat bran increased the content of bioactive compounds. The bioactive compounds from BB- and BC-enriched pastes mainly contributed to their in vitro antioxidant activities. Furthermore, the enrichment of 25% blueberry and 25% blackcurrant powder into oat bran was chosen to investigate the intracellular antioxidant activities in LPS-stimulated RAW264.7 macrophages. The results indicated that the extracts of intestinal digested ABB_25_ and ABC_25_ (bioactive compounds in the extracts) prevented the macrophages from experiencing LPS-stimulated intracellular ROS accumulation, mainly via the Keap1/Nrf2 signalling pathway, enhancing the accumulation of the Nrf2 gene and consequently inducing the expression of the antioxidant genes HO-1, in response to LPS stimulation. In addition, after experiencing the digestion, the extract from the 25% blackcurrant powder enriched paste still exhibited stronger protective effects of in vitro and intracellular antioxidant activity than that of the 25% blueberry powder enriched paste. For future work, a combination of a cell line study with an in vivo study in a mouse model should be developed. We recommend this study focusses on the production of the metabolites of polyphenols that are dominant in the circulation. Taken together, the findings in this study signify the importance of the intake of polyphenol-rich cereal food products.

## Figures and Tables

**Figure 1 antioxidants-10-00388-f001:**
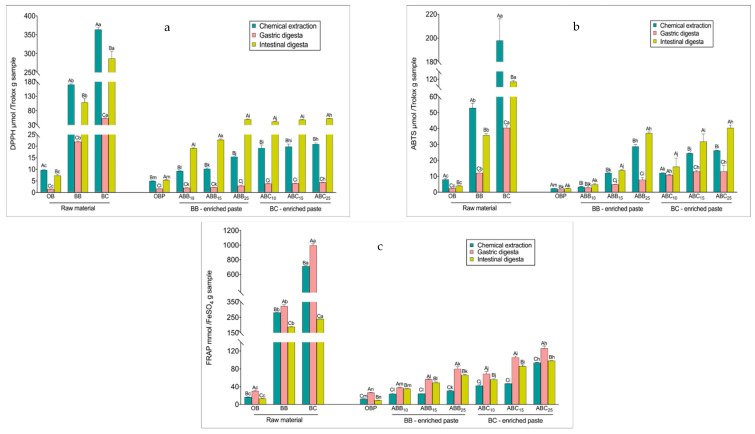
Changes in antioxidant activities during in vitro digestion. DPPH (2-Diphenyl-1-picrylhydrazyl) values (**a**), 2,2′-Azino-bis (3-ethylbenzothiazoline-6-sulfonic acid) diammonium salt (ABTS) values (**b**), and Ferric Reducing Antioxidant Power (FRAP) values (**c**), respectively. Values with different uppercase letters at different digestion phases in one sample are statistically different, while values of bars in the same colour with different lowercase letters are statistically different (*p* < 0.05). OBP = pure oat bran paste; ABB_10_, ABB_15_ and ABB_25_ = oat bran paste enriched with 10, 15 and 25% blueberry powder, respectively; ABC_10_, ABC_15_ and ABC_25_ = oat bran paste enriched with 10, 15 and 25% blackcurrant powder, respectively.

**Figure 2 antioxidants-10-00388-f002:**
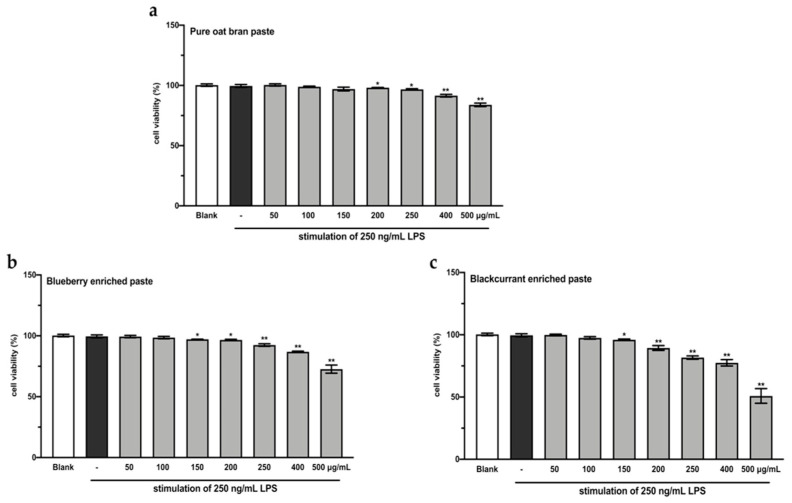
RAW264.7 cell viability after treating with varying concentrations of extracts from digested oat bran paste (**a**), 25% blueberry-enriched paste (**b**), and 25% blackcurrant-enriched paste (**c**) for 48 h. The values represent the mean ± SD of three independent experiments. * *p* < 0.05, ** *p* < 0.01 compared to the control group.

**Figure 3 antioxidants-10-00388-f003:**
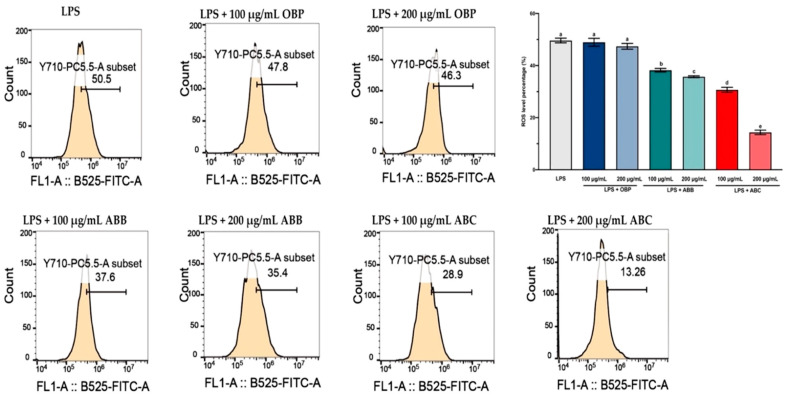
The reactive oxygen species (ROS) levels in LPS-stimulated RAW264.7 macrophages treated with or without the extracts of OBP, ABB, and ABC. The values are represented by mean ± SD. Values with different letters indicate significant differences (*p* < 0.05, *n* = 3). OBP = pure oat bran paste; ABB = 25% of blueberry powder enriched oat bran paste; ABC= 25% of blackcurrant powder enriched oat bran paste.

**Figure 4 antioxidants-10-00388-f004:**
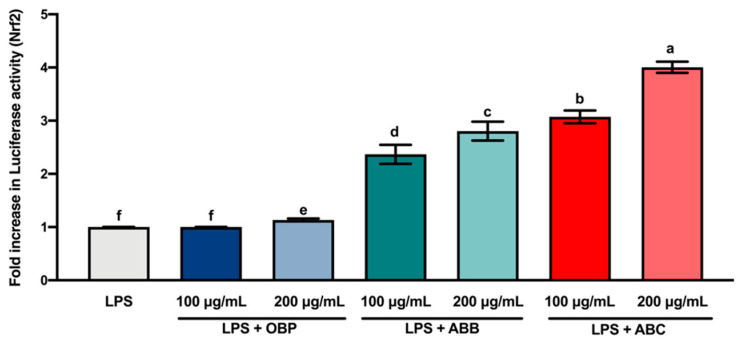
The Luciferase activity (Nrf2) of lipopolysaccharide (LPS)-stimulated RAW264.7 macrophages treated with or without the extracts of OBP, ABB, and ABC. The values are represented by mean ± SD. Values with different letters indicate statistically significant differences (*p* < 0.05, *n* = 3). OBP = pure oat bran paste; ABB = 25% of blueberry powder enriched oat bran paste; ABC = 25% of blackcurrant powder enriched oat bran paste.

**Figure 5 antioxidants-10-00388-f005:**
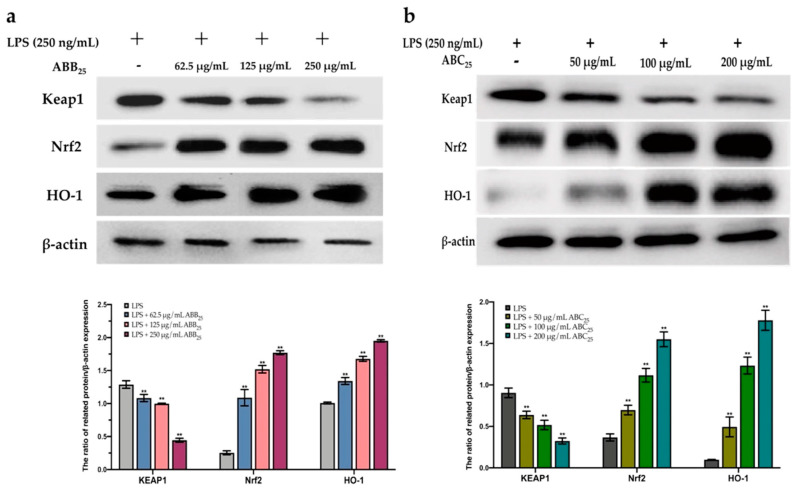
The protein expression of KEAP1, Nrf2, and HO-1 of LPS-stimulated RAW264.7 macrophages treated with or without the varying concentrations of the extracts of ABB_25_ (**a**) and ABC_25_ (**b**). The values are represented by mean ± SD, *n* = 3. Bars with ** mean there is difference from lipopolysaccharide (LPS) group (** *p* < 0.01). ABB_25_ = 25% of blueberry powder enriched oat bran paste; ABC_25_ = 25% of blackcurrant powder enriched oat bran paste.

**Table 1 antioxidants-10-00388-t001:** The total phenol content (TPC) and total flavonoid content (TFC) of raw materials and pastes.

Group	In Vitro Digestion Phase			Total Variation		
	Extraction	Gastric	Intestinal	Samples (A)	Digestion Phase (B)	A*B
Raw material				6.83% (*p < 0.001*)	86.10% (*p < 0.001*)	7.03% (*p < 0.001*)
OB	1.49 ± 0.01 ^Cc^	5.02 ± 0.02 ^Ac^	2.26 ± 0.0 4 ^Bc^			
BB	84.91 ± 2.63 ^Cb^	103.17 ± 1.15 ^Ab^	91.23 ± 0.54 ^Bb^			
BC	97.15 ± 5.31 ^Ca^	186.70 ± 0.23 ^Aa^	122.78 ± 0.67 ^Ba^			
Paste				28.50% (*p < 0.001*)	65.20% (*p < 0.001*)	6.14% (*p < 0.001*)
OBP	0.43 ± 0.01 ^Cm^	4.26 ± 0.00 ^Am^	2.25 ± 0.09 ^Bn^			
ABB_10_	0.53 ± 0.07 ^Cm^	5.22 ± 0.14 ^Al^	3.53 ± 0.05 ^Bm^			
ABB_15_	0.74 ± 0.06 ^Cl^	5.24 ± 0.14 ^Al^	3.92 ± 0.02 ^Bl^			
ABB_25_	1.60 ± 0.06 ^Ck^	6.21 ± 0.23 ^Ak^	4.98 ± 0.09 ^Bk^			
ABC_10_	2.93 ± 0.17 ^Cj^	14.08 ± 0.65 ^Aj^	7.52 ± 0.27 ^Bj^			
ABC_15_	5.85 ± 0.18 ^Ci^	15.89 ± 0.40 ^Ai^	10.44 ± 0.31 ^Bi^			
ABC_25_	10.82 ± 0.36 ^Ch^	18.06 ± 0.19 ^Ah^	11.96 ± 0.08 ^Bh^			
	**In Vitro Digestion Phase**			**Total Variation**		
	**Extraction**	**Gastric**	**Intestinal**	**Samples (A)**	**Digestion Phase (B)**	**A*B**
Raw material				18.20% (*p* < 0.001)	74.70% (*p* < 0.001)	6.80% (*p* < 0.001)
OB	2.82 ± 0.42 ^Bc^	4.77 ± 0.51 ^Ac^	ND			
BB	48.25 ± 2.31 ^Ba^	65.48 ± 1.23 ^Aa^	27.77 ± 2.39 ^Ca^			
BC	36.51 ± 0.93 ^Bb^	43.32 ± 1.22 ^Ab^	18.84 ± 1.40 ^Cb^			
Paste				32.30% (*p* < 0.001)	52.50% (*p* < 0.001)	13.50% (*p* < 0.001)
OBP	0.96 ± 0.07 ^Bm^	2.00 ± 0.19 ^An^	ND			
ABB_10_	5.82 ± 0.39 ^Bk^	10.07 ± 0.47 ^Al^	2.13 ± 0.09 ^Cl^			
ABB_15_	11.17 ± 0.90 ^Bi^	21.27 ± 0.87 ^Aj^	5.26 ± 0.06 ^Cj^			
ABB_25_	24.51 ± 6.15 ^Bh^	37.62 ± 2.71 ^Ah^	7.66 ± 0.19 ^Ch^			
ABC_10_	2.46 ± 0.10 ^Bl^	7.95 ± 0.08 ^Am^	1.55 ± 0.26 ^Cm^			
ABC_15_	7.29 ± 0.76 ^Bj^	17.04 ± 0.44 ^Ak^	3.10 ± 0.50 ^Ck^			
ABC_25_	16.89 ± 2.07 ^Bh^	27.95 ± 0.61 ^Ai^	6.94 ± 0.08 ^Ci^			

Mean ± SD are presented (*n* = 3). Raw materials and pastes were compared separately. Comparison within the same raw material is expressed by uppercase letters, while comparison within the same column is expressed by lowercase letters. Values with different letters, per sample, are statistically different (*p* < 0.05). Abbreviations: TPC = total phenolic content; TFC = total flavonoid content; OB = oat bran; BB = blueberry powder; BC = blackcurrant powder; OBP = pure oat bran paste; ABB_10_, ABB_15_ and ABB_25_ = oat bran paste enriched with 10, 15 and 25% blueberry powder, respectively; ABC_10_, ABC_15_ and ABC_25_ = oat bran paste enriched with 10, 15 and 25% blackcurrant powder, respectively. All values are based on dry basis.

**Table 2 antioxidants-10-00388-t002:** The changes in TMAC values of the extracts during in vitro digestion (mg/100 g sample).

TMAC (mg Cy-3GE/100 g Sample)			
Group	In Vitro Digestion Phase		
	Extraction	Gastric	Intestinal
Raw material			
Oat bran	0.03 ± 0.01 ^Ac^	ND	ND
Blueberry powder	14.96 ± 1.48 ^Ab^	0.99 ± 0.03 ^Bb^	0.62 ± 0.02 ^Cb^
Blackcurrant powder	36.27 ± 1.69 ^Aa^	3.90 ± 0.03 ^Ba^	3.43 ± 0.04 ^Ca^
Paste			
OBP	ND	ND	ND
ABB_10_	0.25 ± 0.02 ^Al^	0.11 ± 0.02 ^Bk^	ND
ABB_15_	0.30 ± 0.01 ^Ak^	0.13 ± 0.02 ^Bk^	0.04 ± 0.01 ^Ck^
ABB_25_	0.41 ± 0.00 ^Aj^	0.16 ± 0.02 ^Bj^	0.08 ± 0.00 ^Cj^
ABC_10_	0.40 ± 0.01 ^Ai^	0.18 ± 0.03 ^Bj^	0.12 ± 0.01 ^Ci^
ABC_15_	0.64 ± 0.00 ^Ah^	0.40 ± 0.03 ^Bi^	0.13 ± 0.02 ^Ci^
ABC_25_	0.66 ± 0.03 ^Ah^	0.62 ± 0.02 ^Bh^	0.35 ± 0.01 ^Ch^

Values are mean ± standard deviation, *n* = 3. Raw materials and pastes were compared separately. Values with different uppercase letters, in the same row, are statistically different (*p < 0.05*), while values with different lowercase letters, in the same column are statistically different (*p < 0.05*). TMAC = total monomeric anthocyanins content; OBP = pure oat bran paste; ABB_10_, ABB_15_ and ABB_25_ = oat bran paste enriched with 10, 15 and 25% blueberry powder, respectively; ABC_10_, ABC_15_ and ABC_25_ = oat bran paste enriched with 10, 15 and 25% blackcurrant powder, respectively; ND = no data. All values are based on the dry basis.

**Table 3 antioxidants-10-00388-t003:** Pearson’s correlation between phenolic compounds and three antioxidant assays values.

	Before Digestion	Gastric	Intestinal
**Pearson’s Correlation between TPC and Three Antioxidant Assays Values**			
**DPPH values**	0.8708 **	0.9783 **	0.8641 **
**ABTS values**	0.6867 **	0.7748 **	0.6941 **
**FRAP values**	0.8449 **	0.9382 **	0.9279 **
**Pearson’s Correlation between TFC and Three Antioxidant Assays Values**			
**DPPH values**	0.5724 *	0.3737	0.4996 *
**ABTS values**	0.4333 *	0.2707	0.4245 *
**FRAP values**	0.5080 *	0.3279	0.7968 **
**Pearson’s Correlation between TMAC and Three Antioxidant Assays Values**			
**DPPH values**	0.9922 **	0.9820 **	0.9790 **
**ABTS values**	0.9551 **	0.9650 **	0.9188 **
**FRAP values**	0.9951 **	0.9962 **	0.8376 **

* *p* < 0.05, ** *p* < 0.01, *n* = 3. TPC = total phenolic content; TFC = total flavonoid content; TMAC = total monomeric anthocyanins content.

## Data Availability

The data presented in this study are available on request from the corresponding author.

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
