# Peer review of "The Effects of Bioactive Compounds from Blueberry and Blackcurrant Powder on Oat Bran Pastes: Enhancing In Vitro Antioxidant Activity and Reducing Reactive Oxygen Species in Lipopolysaccharide-Stimulated Raw264.7 Macrophages"

_antioxidants, 2021, doi:10.3390/antiox10030388_

Round 1
Reviewer 1 Report
1. Line 28, "weres" should be "were".
2. Line 61-64, what is the reference of this sentence.
3. Line 64-65, it mentioned ’’It has been reported that blueberry and blackcurrant fruits contain a large number of polyphenols, with high antioxidant properties’’. However, it did not specify the types of phenolic compounds in blueberry and blackcurrant.
4. Line 64-66, mention the sentence of blueberries and blackcurrants cannot match the context here.
5. Line 270, "thus" should be "Thus".
6. Lines 285-288, the author of the reference should be written at the beginning of this sentence.
7. Line 324, there should be space between Pg and before.
8. Line 421, table 3 should be table 4.
9. Line 333-339, in the description of Table 2, there is no mention of how many samples for mean. Besides, it also not describes how to calculate the value of (A) and (B) in Table 2.
10. Line 428-433, should put the reference of previous research in the text.
11. In introduction should mention more about why blueberries and blackcurrants was chosen.
12. Table 2. showed the TFC and TPC would increase after gastric digestion phase and decrease after intestinal digestion phase, and Table 3. showed the anthocyanidin content would decrease after digestion phase. The results didn’t meet the Figure 1 result, please explain the reason.
Reviewer 2 Report
The manuscript “The synergistic effects of bioactive compounds from blueberry and blackcurrant powder on oat bran pastes: enhancing the in vitro antioxidant activity and reducing the intracellular reactive oxygen species in LPS-stimulated RAW264.7 via nuclear factor erythroid 2-related factor 2 (Nrf2) signalling pathway” by Hui et al. studied the influence of mixing oat bran with blueberry and blackcurrant on the variation of the phenolic compounds, flavonoids and total antioxidant capacity during digestion. The authors measured as well the in vitro antioxidants effects on RAW264.7 macrophages.
The manuscript presents a large number of results; however, several errors have been detected. Some comments are listed below:
- The authors should reduce their title length and avoid over-interpretation of results (they did not evaluate synergistic effects).
- In the abstract, the authors should eliminate the first sentence.
- Line 41, oxidative stress is not overexpressed. Rewrite the sentence.
- In line 125, the authors should explain why they used a “chemical extraction” it is not clear when they presented it.
- Line 133, r/min?
- Section 2.7. The authors affirm they identify and quantify individual anthocyanin by the pH differential method. This is a severe mistake. The pH differential method cannot distinguish among anthocyanins. The different wavelengths and coefficient s of molar absorptivity are useful for selecting the best conditions to quantify total monomeric anthocyanins by using the most appropriate (the most representative) standard. The anthocyanins spectrum are overlapped, and the maximum absorbance peak only differs on 10 nm approximately. The authors should eliminate this section (3.2.) and reformulate it again, considering the total monomeric anthocyanin value.
- In sections 3.1. and 3.3. the authors found different results that were associated with the difference in the pH of the medium. I wonder if these differences are produced by an increase/decrease in the content of phenolics or if, on the contrary, they are results of an experimental error due to the lack of pH control of samples once dissolved from the lyophilized digested media. pH has an essential role in spectrophotometric assays. The authors should have considered controlling it. It is weird to find increases in TPC, TFC, and FRAP while decreasing anthocyanins, DPPH, and ABTS.
- Then, the authors calculated Pearson’s correlation coefficients. Evidently, if they calculate them, including all digestion phases, non-significant correlations would be found. Instead, they calculated individually in each phase. Here is where I also think the effect of digestion media pH and other interferences is giving erroneous results leading the authors to extract some wrong conclusions.
- On the cellular assays, the authors should have measured blueberry and blackcurrant’s effect without oat bran to comprehend why oat bran’s impact adds to the berries’ effect.
- In Figure 5A, I wonder if the authors have higher quality pictures of their WB. I do not think these bands are representative of the obtained results.
- In general, I find the cellular experiments are well-conducted; however, there are not enough to support the whole manuscript. On the other hand, the in vitro digestion experiments present several flaws. There is no cohesion among parts. Even the conclusion section did not mention the digestion part.
Hence, this manuscript is not ready to be published in Antioxidants.
Reviewer 3 Report
The paper has a lot of lab work. Basically it could be said that the oat bran is enriched with 3 different proportions (incorporation of 10%, 15% and 25% of black currant or blueberry podwer). These samples undergo a digestive process.
In the first part of the article (up to section 3.5.) total polyphenols, flavonoids and anthocyanin’s are analyzed as well as antiradical activity (by DPPH, ABTS and FRAP) both in the original samples (separately) and in the pastes generated with oats; and also, after the samples have followed a simulation of digestion "in vitro" with two stages, gastric and intestinal.
Tables 2, 3 and 4, figure 1 as well as sections 3.1. to 3.4. they work and explain all these results which represents more than 75% of the article.
A complete statistical analysis is done. But the big problem is that all these methods are semi-quantitative, that is, they serve to give an idea, but they are not valid to reach the conclusions raised in the article. In addition, the influence of pH on post-digestive processes in the mentioned methods should be correctly analyzed, since the mentioned radical scavenging methods are pH-dependent.
On the other hand, the proposed quantification of anthocyanins is not chemically correct, quantitatively (which would allow us to carry out the proposed statistical study) but only for guidance. Since, can Cyanidin be differentiated from Peonidin, when there is only one reading of the Absorbance with minimal variation of λ (510.5 vs 511 nm)?
The same with Delphinidin and Malvidin (can you differentiate the absorbance measured at 522.5 versus 520 nm?)
The answer is no. The appropriate methodology would be a quantification of the polyphenolic compounds, of the commented families (anthocyanin’s, flavonoids or any other polyphenol), as well as the avenanthramides (specific to oats) through HPLC, seeking the most appropriate separation methodology.
Pearson's correlations of radical scavenging activities with flavonoids and anthocyanins do not make much sense, since antiradical activities are signified to all included compounds while flavonoids and anthocyanins are a subset of them. All possible synergistic effects or interactions that increase radical scavenging activity are left out. The fact that the correlations with anthocyanin’s are so high is mainly due to the fact that they are not well determined.
In addition to the methodological aspects exposed, in the introduction and in material and methods there are many aspects that can be improved. Various questions and issues for authors have been marked and set in the pdf document.
In the last part of the article (25%) a CCK-8 test was carried out to evaluate the effects of intestinally digested blueberry and blackcurrant enriched pasta extracts on RAW264.7 cell viability. It works at different concentrations of extracts from 50 to 500 µg/mL. The authors indicate significant differences from 150-200 µg/mL (although the graphs make it impossible to see that the above is correct). In any case, at the highest concentration tested, there appear to be significant differences.
The decrease in ROS, especially in the sample containing blackcurrant powder (BC) is really remarkable and important.
Surely, a good article could be proposed with a quantification of the main antioxidant compounds (by HPLC) and explaining (in an extended way) the second part of the article (from section 3.5.)
Other aspects to correct:
- The CCK-8 test is done only with the digested samples, so the increase due to the “in vitro” digestion process cannot be analyzed.
- All the abbreviations, although they are well known, they must be put with the full name the first time
- There are many reagents that are not reflected in section 2.1.
- The presentation of the tables is not friendly. In fact, some of the results are not understood

Reviewer 4 Report
Manuscript number Antioxidants-1106938
entitled: The synergistic effects of bioactive compounds from blueberry and blackcurrant powder on oat bran pastes: enhancing the in vitro antioxidant activity and reducing the intracellular reactive oxygen species in LPS-stimulated RAW264.7 via nuclear factor erythroid 2-related factor 2 (Nrf2) signalling pathway
This is valuable and a well-conducted scientific study, done thoroughly and expressed concisely. Therefore, the manuscript is suitable for Antioxidants after considering the below comments:
- Please provide chemical structures of main chemicals which contain the blueberry and blackcurrant powder, and present in Table 1 (page 4) anthocyanidins. Please change the capital letter “N” present in the word “Anthocyanidins” in the title of Table 1.
- Please explain the term avenanthramides line 78 page 2, and please provide chemical structures of main chemicals from this group, and maybe brief information about this class of chemicals.
- Please check the title; possible should be “signaling” not “signalling”..
Round 2
Reviewer 1 Report
The citation format is not uniform.
Author Response
Thank u for your comment and suggestion. We have carefully checked the reference style in our revised MS, and kept them consistency.
Reviewer 2 Report
The authors have correctly answered the four reviewers’ comments. Even if the article may present several flaws, the authors have corrected them and explained their errors. Fuerthermore, they added explanations on the MS for the article’s future readers.
I consider it could be accepted.
Author Response
Thank you for your constructive comments.
Reviewer 3 Report
There are several points that do not answer. For example, the influence that pH may have on the determination of the methods. I did a lot of suggestions in the pdf and I saw that some of them (for instance, to say where you bought all the materials) has not been take into account
Do the high correlations between anthocyanins and radical scavenging activity, according to you, indicate that antiradical activity increases practically linearly when anthocyanins do? Is it not that since they are all electron transfer methods and all of them are measured by absorbance, the correlation is given by the similarity of the methodology? Have you used the ORAC method, which measures proton transfer? Maybe you should.
You say that your study aimed to demonstrate that the addition of berry powders into oat bran significantly improved the bioactive compounds and antioxidant capacity of the pastes. Honestly, if I add a compound (with antioxidant capacity) to another that does not have it (or is very mild), it is evident that it will increase the antioxidant capacity. My question is, what compounds increase it? (not the family, but the concrete names) which ones remain after the digestive simulation? What influence do these compounds have on antioxidant activity?
Without serious analysis by HPLC this article cannot be accepted. You comment that this analysis has already been sent to another paper. My suggestion would be that you try to integrate everything into one article with enough chemical determination to be considered in Antioxidants. If they have already done HPLC, you know the accuracy of the method compared to those they have used and of the need to incorporate it to make it a serious article.
On the other hand, I have not seen many of the suggestions that I made in the article itself changed. I have not reviewed all of them, but I have seen that, for example, they have not added all the products in materials. I imagine there will be others.
I insist that the last part of the article that incorporates the study with RAW264.7 seems good to me, but insufficient for an article. And the whole part of radical scavenging activity is very justified and explained, but the methods themselves do not serve to draw those conclusions
